# Construction and Application of Reservoir Flood Control Operation Rules Using the Decision Tree Algorithm

Yanfang Diao [1], Chengmin Wang [2], Hao Wang [1] and Yanli Liu [3,4,*]

1  College of Water Conservancy and Civil Engineering, Shandong Agricultural University, Tai'an 271018, China; diaoyanfang@sdau.edu.cn (Y.D.); wh17861509336@163.com (H.W.)
2  Rizhao Reservoir Management and Operation Center, Rizhao 276816, China; sdwcm2006@163.com
3  State Key Laboratory of Hydrology-Water Resources and Hydraulic Engineering, Nanjing Hydraulic Research Institute, Nanjing 210029, China
4  Yangtze Institute for Conservation and Development, Nanjing 210098, China
*  Correspondence: ylliu@nhri.cn

**Abstract:** Current conventional and optimal reservoir flood control operation methods insufficiently utilize historical reservoir operation data, which include rainfall, runoff generation, and inflow from the watershed, as well as the operational experience of decision makers over many years. Therefore, this study proposed and evaluated a new method for extracting reservoir flood control operation rules from historical operation data using the C4.5 algorithm. Thus, in this paper, the C4.5 algorithm is first introduced; then, the generation of the flood control operation dataset, the construction of decision tree-based (DT-based) rules, and the subsequent design of a real-time operating scheme are detailed. A case study of the Rizhao Reservoir is then employed to demonstrate the feasibility and even superiority of the operating scheme formulated using DT-based rules. Compared with previously proposed conventional and optimal reservoir operation methods, the DT-based method has the advantages of strong and convenient adaptability, enabling decision makers to effectively guide real-time reservoir operation.

**Keywords:** flood control operation; C4.5 algorithm; DT-based rules; nodes; discharge

## 1. Introduction

Flood disasters are currently among the major global problems faced by human society. From 1989 to 2018, 3945 major flood disasters occurred around the world, with China, India, the United States, and Indonesia experiencing the largest number: about 1200 in total [1]. There were 109 flood disasters worldwide in 2018, causing 1995 deaths, affecting 12.62 million people, and resulting in $4.5 billion in direct economic losses [2]. Although global flood deaths and affected populations have shown a continuous downward trend over the past 30 years, economic losses have shown an upward trend. Owing to the frequency of and significant economic losses associated with flood disasters, a considerable number of water conservation projects have been undertaken to reduce the adverse effects of floods. Among these, reservoirs are created by constructing a dam across a river. However, with ongoing socioeconomic development, the purpose of the reservoir has expanded from guaranteeing flood control safety of the river to including the provision of power generation, water supply, irrigation, ecological environment maintenance, navigation, sediment control, recreation, fisheries, etc. As of April 2020, 58,713 large dams have been constructed worldwide [3]. According to a report by the World Commission on Dams, the improvement in operation and maintenance of existing dams offers opportunities to address local or regional development and to minimize social and environmental impacts [4]. To do so, it is necessary to implement a scientific and reasonable reservoir flood control operation strategy. Thus, the goal of reservoir flood control operation studies is to define an optimal operation policy for a given reservoir that balances its various

purposes [5]. This policy represents a powerful tool for the guidance of reservoir operation, serving as not only a decision-making reference during the planning and design of a water conservancy project but also a key to realizing the comprehensive benefits of the reservoir during its operation.

Existing reservoir flood control operation methods can be roughly divided into two groups: conventional operation methods and algorithmically optimized operation methods. Conventional reservoir operation methods are semi-empirical and semi-theoretical and are presented in the form of flood control operation graphs or tables. During real-time flood control operation, operational decisions (e.g., reservoir discharge or hydroelectric power generation) during each period are specified as a function of the appropriate available information (e.g., the current or previous reservoir water level, current or previous reservoir inflow, and time of year) [6]. Such conventional methods have been widely used in reservoir operations owing to their intuitive and practical structure; however, they often fail to consider the latest operational data or to address the complex nonlinearity that exists between the relevant dependent and independent variables [7]. Thus, as the complexity and interdependency of the systems considered in reservoir management increase, it becomes more difficult to obtain an optimal operating scheme using conventional methods.

Optimization algorithms have therefore been increasingly applied to formulate reservoir flood control operation strategies, effectively addressing the shortcomings of conventional methods [8]. In the past decades, a wide range of optimization algorithms have been proposed that can generally be classified into conventional optimization algorithms and heuristic intelligent algorithms. Conventional optimization algorithms include linear programming [9–11], nonlinear programming, dynamic programming (DP) [12–14], and progressive optimality algorithm (POA) [15,16] approaches as well as various improvements thereof, such as the multi-stage DP [17,18], incremental DP [19,20], stochastic DP [21], parallel DP [22], and DP combined with POA (DP–POA) method [23]. However, when faced with a sufficiently complex flood control system composed of multiple reservoirs, flood storage and detention areas, lakes, and other infrastructure, conventional optimization algorithms have obvious limitations, including a low convergence efficiency and the "curse of dimensionality". For example, as the number of reservoirs increases, the computational scale of DP increases exponentially. To address such issues, modern computing technology has enabled the development of heuristic intelligent algorithms based on artificial intelligence, resulting in general-purpose stochastic search methods that simulate natural selection and biological evolution. As they can be directly applied to address complex problems with nonlinear, discontinuous, non-differentiable, and multi-dimensional characteristics, they have been widely used to optimize flood control operations. At present, the most common heuristic intelligent algorithms include the genetic algorithm [24,25], non-dominated sorting genetic algorithm [26–28], particle swarm optimization [29,30], ant colony optimization [31,32], artificial neural network [33,34], support vector machine [35], simulated annealing [36], immune-inspired optimization [37], evolutionary algorithm [38,39], cultured evolutionary algorithm [40,41], and honey-bee mating optimization algorithm [42]. However, although heuristic intelligent algorithms can determine an optimal operating policy, there remain many problems in their practical application. The flaws intrinsic to most heuristic intelligent algorithms include premature convergence owing to local fast convergence, poor local search capability owing to a large number of global searches, and long iteration time [43]. Furthermore, the solutions provided by most algorithms are limited by the available calculation time as well as the constraints associated with certain optimizations [44].

At present, both flood control operation methods insufficiently utilize historical reservoir operation data. Importantly, these data contain not only the characteristics of and laws describing runoff generation and inflow from the watershed but also the vast experience of reservoir managers, which provides information supporting operating decisions according to different inflow scenarios. As a considerable body of reservoir operation data has been accumulated by water conservation departments, the use of data mining

technology to extract flood control operation rules from these data offers a new method for real-time reservoir flood control operation. The decision tree (DT) algorithm is the most commonly used data mining model, as it creates decision rules and classification results following a tree structure [45]. The DT algorithm has the advantages of being easier to understand, being easier to implement, and requiring relatively less workload than other approaches. Therefore, it has been widely used to address water conservation problems such as flood forecasting [46,47], flood or drought risk assessment [48–53], flood or drought classification [54,55], water quality prediction [56,57], inter-basin water transfer dispatching [58], water level prediction [59,60], and hydropower station power generation dispatching [61]. Noymanee and Theeramunkong [46] adopted the boosted decision tree regression to forecast flood water levels in a real-time manner and achieved high forecasting accuracy. Nafari et al. [49] confirmed that the tree-based models were more accurate than the stage-damage function from Australia in a flood risk assessment. Sikorska et al. [54] presented a flood classification for identifying flood patterns at a catchment scale by means of a fuzzy decision tree, and the results showed that this method bore additional potential for analyses of flood patterns. Xi et al. [58] used the decision tree method to determine the diversion amount according to the inter-basin water transfer rules. Parvez et al. [61] proved that the C4.5 algorithm was more feasible for rapidly generating the schedules of cascaded hydropower plants. However, it has rarely been used in reservoir flood control operations. This study aims to formulate reservoir flood control operation rules using the DT algorithm.

The remainder of this paper is organized as follows: Section 2.1 describes the DT algorithm, Section 2.2 presents the construction of flood control operation rules using the DT algorithm (DT-based rules), and Section 2.3 designs a real-time operation procedure using these DT-based rules; Section 3 then introduces and discusses the results of a case study application of the proposed DT algorithm; and Section 4 provides a summary of the conclusions.

## 2. Proposed Method

### 2.1. DT Algorithm

The DT algorithm, first proposed in the 1960s, is a greedy local search algorithm that first analyzes and processes historical data to construct a DT through top-down induction and then uses the DT to analyze new data [45]. The DT algorithm is usually constructed beginning at the top of the tree and proceeding down into the branches, each of which represents a decision, and then into the leaves (or nodes), each of which is assigned a classifier value. Typical DT algorithms include the iterative dichotomiser 3 (ID3) [62], C4.5 [63], and classification and regression tree (CART) [64].

The ID3 algorithm uses the information gain as the splitting criterion for the DT to realize the induction and classification of the data. This has the advantages of providing a concise and clear basic theory as well as a strong learning ability. However, the ID3 algorithm has several drawbacks: it does not consider numerical attributes, missing attribute values are not taken into account, no pruning process is included, and it does not handle data with high dimensionality [65]. The C4.5 algorithm is an enhanced version of the ID3 algorithm that applies the information gain ratio rather than information gain itself as the standard for attribute selection. This addresses several of the shortcomings of the ID3 algorithm by realizing numerical attribute treatment, working with missing values, and introducing a pruning process [65]. The CART algorithm employs a binary induction method; that is, the DT generated by this algorithm is in the form of a binary tree. However, the C4.5 algorithm can handle continuous attributes more effectively than the CART algorithm. Therefore, the C4.5 algorithm was used to construct the DT-based flood control operation rules in this study. The specific steps of this algorithm are as follows.

Suppose that $p$ is the number of samples in the set $S$; the class label attribute $P_i$ has $c$ different values, where $P_i$ ($i = 1, 2, \ldots , c$); and $p_i$ is the number of samples in class $P_i$. The information entropy of $S$ is thus given by

$$I(S) = -\sum_{i=1}^{c} (p_i/p) \log_2 (p_i/p) \tag{1}$$

Suppose that the set attribute $A$ has $m$ different values $\{a_1, a_2, \ldots , a_m\}$. Set $S$ can thus be divided into $m$ subsets $\{U_1, U_2, \ldots , U_m\}$, where $U_j$ contains a number of values from $S$ in the sample and it has a value of $a_j$ in $A$. $U_j$ assumes that $p_{ij}$ is a subset of the samples of class $P_i$. Thus, the entropy or expected information of the subsets divided by $A$ is

$$E(A) = \sum_{j=1}^{m} \left[ \frac{p_{1j} + p_{2j} + \cdots + p_{cj}}{p} I(S) \right] \tag{2}$$

where $(p_{1j} + p_{2j} + \cdots + p_{cj})/p$ is the weight of the $j$th subset. The smaller the entropy value, the higher the purity of the subset.

To determine whether the selected attribute $A$ can effectively reduce the overall entropy, the information gain of attribute $A$ can be defined as follows:

$$G_{\text{ain}}(A) = I(S) - E(A) \tag{3}$$

in which a higher $G_{ain}(A)$ indicates a larger reduction in entropy and therefore a better attribute. Suppose that $R(A)$ is the information gain ratio, defined as the ratio of $G_{ain}(A)$ to the split information $S_{\text{pliti}}(A) = -\sum_{j=1}^{m} (p_{ij}/|p_j|) \log_2 (p_{ij}/|p_j|)$, namely,

$$R(A) = G_{\text{ain}}(A)/S_{\text{pliti}}(A) \tag{4}$$

Calculating the $R(A)$ of each attribute and attribute dataset using Equation (4), the attribute with the largest $R(A)$ is taken as the split attribute to create nodes and to divide branch samples until all samples under a given node belong to the same category or the attribute can no longer be divided. Finally, reservoir flood control operation rules are constructed based on the derived DT (DT-based rules) by considering the various factors influencing reservoir discharge.

### 2.2. Construction of DT-Based Flood Control Operation Rules

2.2.1. Generation of Reservoir Flood Control Operation Dataset

The main factors affecting reservoir discharge include flood occurrence time, rainfall, net rainfall, reservoir water level, rate of inflow, and volume of inflow. In regions with an uneven distribution of rainfall throughout the year, the rainfall volume and intensity in the flood season are usually high, resulting in large discharges, whereas there is little rain in the non-flood season, resulting in small discharges. Therefore, the occurrence of flooding is related to reservoir discharge. The water level reflects the water stored by the reservoir; the higher the water level, the greater the discharge required during flooding to ensure the safety of the dam. The total rainfall, net rainfall, rate of inflow, and volume of inflow reflect the amount and intensity of inflow into the reservoir and are directly proportional to required discharge; that is, the larger these parameters, the greater the required discharge, and vice versa. This analysis indicates that a reservoir flood control operation dataset includes many attributes, among which the discharge in each period is the decision attribute, whereas the others are conditional attributes.

2.2.2. Construction of DT-Based Rules

To construct the reservoir operation rules using the C4.5 algorithm, all floods in the dataset are first divided into training and verification samples. Then, the attribute values for each training sample period are calculated to construct the flood control operation dataset. Finally, the C4.5 algorithm is used to extract operation rules from this dataset.

*2.3. Real-Time Operation Procedure*

Given a certain verification sample, the procedure for real-time operation is as follows:

Step 1: Let $i = 1$.

Step 2: Use the conditional attributes in flood period $i$ as inputs to the DT-based rules to solve the discharge $q_i$ in period $i$, and calculate the water level $z_i$ in period $i$ using the water balance equation.

Step 3: To ensure downstream safety, confirm that $q_i$ is less than $q^*$, the maximum discharge of the verification sample as regulated by conventional operation rules. If this is true, proceed to Step 4; otherwise, set $q_i = q^*$.

Step 4: If $q_i$ is less than $q(z_i)$, which is the reservoir discharge capacity when the water level is $z_i$, proceed to Step 5; otherwise, set $q_i = q(z_i)$.

Step 5: Let $i = i + 1$. If $i < T$, where $T$ is the verification sample period count, return to Step 2; otherwise, end the operation.

The real-time operation procedure is illustrated in Figure 1.

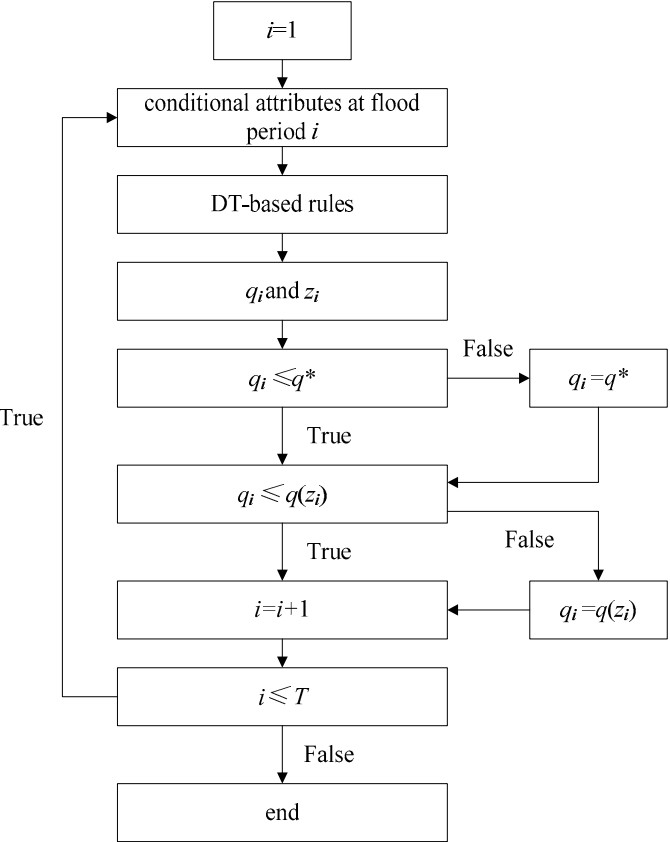

**Figure 1.** The flowchart of the real-time reservoir operation procedure using DT-based rules.

## 3. Case Study

*3.1. Study Area*

The Rizhao Reservoir was used as a case study for the application of the proposed DT-based rule in this study. This reservoir is located 16 km west of Donggang District, Rizhao City in Shandong Province, China, and belongs to the upper and middle reaches of the Futuan River. Construction of the Rizhao Reservoir began in October 1958 and was completed in June 1959. It is a large type-II reservoir with multi-year regulation used mainly to provide flood control and irrigation in combination with aquaculture, power generation, water supply, and other secondary objectives. The climate of the Rizhao Reservoir basin generally exhibits the characteristics of humid and semi-humid regions. Other basic parameters of the reservoir are shown in Table 1.

**Table 1.** Basic parameters of the Rizhao Reservoir.

| Items | Unit | Reservoir |
|---|---|---|
| Catchment area | km$^2$ | 548 |
| Total storage | 10$^8$ m$^3$ | 3.1805 |
| Active storage | 10$^8$ m$^3$ | 1.8232 |
| Design standard | % | 1 |
| Check standard | % | 0.02 |
| Checked flood level | m | 46.51 |
| Designed flood level | m | 44.02 |
| Normal water level | m | 42.5 |
| Flood limited water level | m | 42 |

To ensure safety downstream of the Rizhao Reservoir, two control discharges and two high-volume discharge states have been established: when the water level Z ≤ 43.46 m, the control discharge is 1000 m$^3$/s; when 43.46 m < Z ≤ 43.79 m, the control discharge is 1900 m$^3$/s; when 43.79 m < Z ≤ 44.02 m, the spillway sluices are completely opened; and when Z > 44.02 m, the spillway sluices and the north water release tunnel are completely opened together.

*3.2. Flood Control Operation Dataset for the Rizhao Reservoir*

Based on the historical operation data, 44 floods in 1970, 1974, 1975, 1976, 1998, and 2001–2021 were used as operation data, all of which required discharge by the spillway sluices. As the Rizhao Reservoir has high regulation performance and ensures downstream safety, the flood volume plays a major role in flood operation. In contrast, according to the climate characteristics of the reservoir basin, precipitation is unevenly distributed throughout the year. The flood season lasts from June to September, during which approximately 80% of annual precipitation is received; approximately 60% of annual precipitation is received in July and August alone. The non-flood season lasts from November to April of the following year, during which little precipitation is received. Considering the above characteristics, the flood occurrence time, reservoir water level, cumulative net rainfall, and discharge were taken as the attributes in the flood control operation dataset; the first three of these attributes were defined as conditional attributes and the last attribute was defined as the decision attribute.

*3.3. Construction of DT-Based Rules for the Rizhao Reservoir*

Forty floods occurring in 1970, 1974, 1975, 1976, 1998, and 2001–2018 were included in the training sample while four floods occurring in 2019–2021 were used as verification samples. First, the flood occurrence times, initial water levels, cumulative net rainfall, and discharges in the 40 floods constituting the training sample were sorted and classified for use as the flood control operation dataset. When the information gain rate was at its maximum, the classification of flood occurrence time, initial water level, cumulative net rainfall, and discharge were as shown in Tables 2 and 3. Owing to the uneven distribution of precipitation in the Rizhao Reservoir basin, its hydrological year was divided into three stages: June and September, July to August, and October to May of the following year, as shown in Table 2. Finally, the flood control operation dataset was input into the C4.5 algorithm to generate the DT-based rules for Rizhao Reservoir operation, as shown in Figure 2. It can be seen in Figure 2 that, (1) in the case of the same initial water level and cumulative net rainfall, the discharge in the flood season is greater than that in the non-flood season and, (2) the higher the initial water level and cumulative net rainfall, the greater the discharge.

**Table 2.** Classification of cumulative net rainfall.

| Flood Occurrence Time | Classification/mm | | | |
| --- | --- | --- | --- | --- |
| | Grade 1 | Grade 2 | Grade 3 | Grade 4 |
| June and September | [0,170) | [170,275) | [275,340) | ≥340 |
| July to August | [0,153) | [153,248) | [248,306) | ≥306 |
| October to May of the next year | [0,191) | [191,310) | [310,382) | ≥382 |

**Table 3.** Classification of all attributes.

| Grade | Flood Occurrence Time | Initial Water Level/m | Cumulative Net Rainfall/mm | Discharge/m³/s |
| --- | --- | --- | --- | --- |
| Grade 1 | June and September | ≤$Z_{limit}$ | | ≤100 |
| Grade 2 | July to August | ($Z_{limit}$,42.91] | As show in Table 2 | (100,1000] |
| Grade 3 | October to May of the next year | >42.91 | | (1000,2000] |
| Grade 4 | | | | (2000,2500] |
| Grade 5 | | | | >2500 |

Note: $Z_{limit}$ represents the flood limited water level.

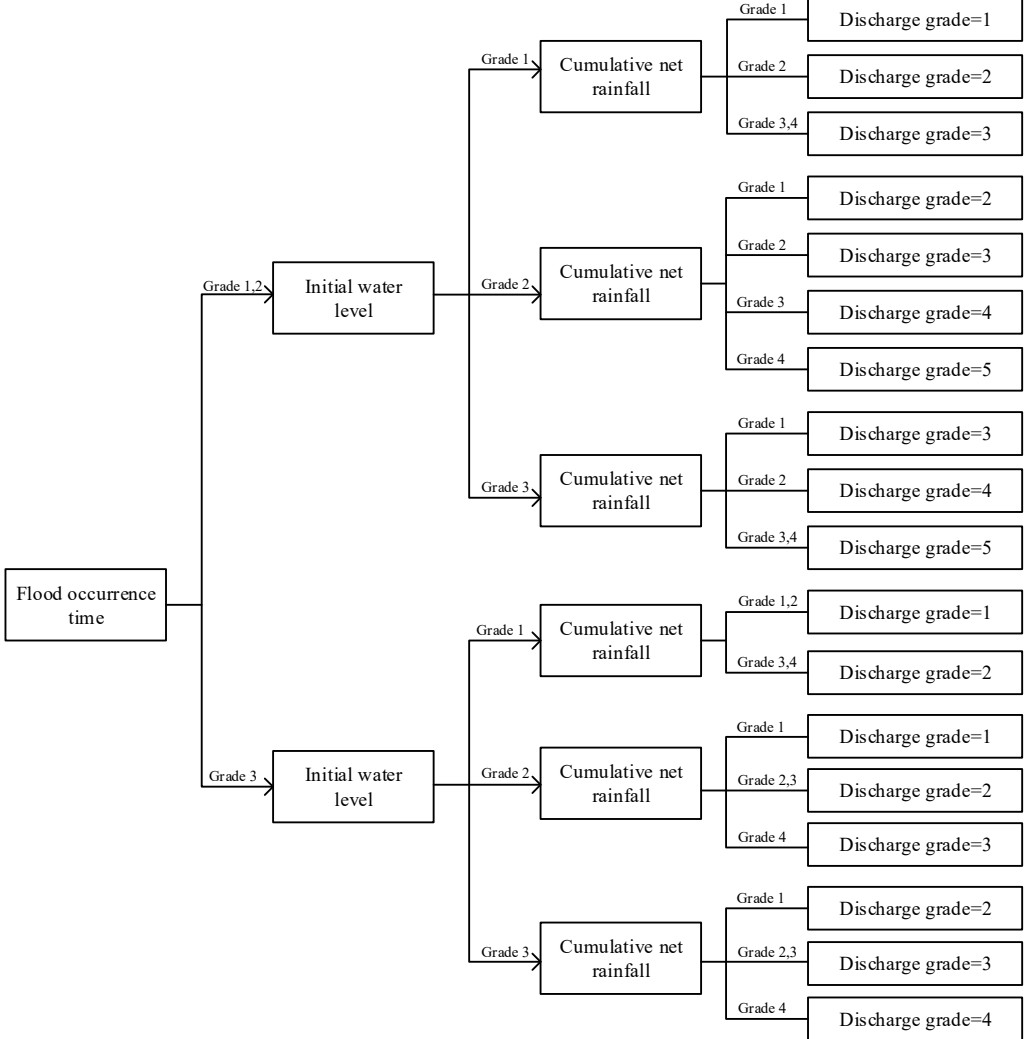

**Figure 2.** DT-based operation rules.

### 3.4. Results and Discussion

The discharges and water levels of the four verification samples were obtained according to the real-time operation procedure depicted in Figure 1 and are plotted together with the measured inflows, discharges, and water levels in Figures 3–6. The maximum discharge and water level are listed in Table 4, in which ① represents flood regulation results obtained using the DT-based rules, ② represents the measured values, and ③ represents the flood regulation results based on the conventional operation rules. The relative errors of the maximum discharge and maximum water level reported in Table 4 were determined, respectively, by the following:

$$r_q = \frac{|q_{\mathrm{maxDT}} - q_{\mathrm{max}}|}{q_{\mathrm{maxDT}}} \times 100\% \tag{5}$$

$$z_q = \frac{|z_{\mathrm{maxDT}} - z_{\mathrm{max}}|}{z_{\mathrm{maxDT}}} \times 100\% \tag{6}$$

where $q_{\mathrm{maxDT}}$ is the maximum discharge according to DT-based rules; $z_{\mathrm{maxDT}}$ is the maximum water level according to DT-based rules; $q_{\mathrm{max}}$ is the measured maximum discharge or the maximum discharge determined by the conventional operation rules; and $z_{\mathrm{max}}$ is the measured maximum water level or the maximum water level determined by the conventional operation rules.

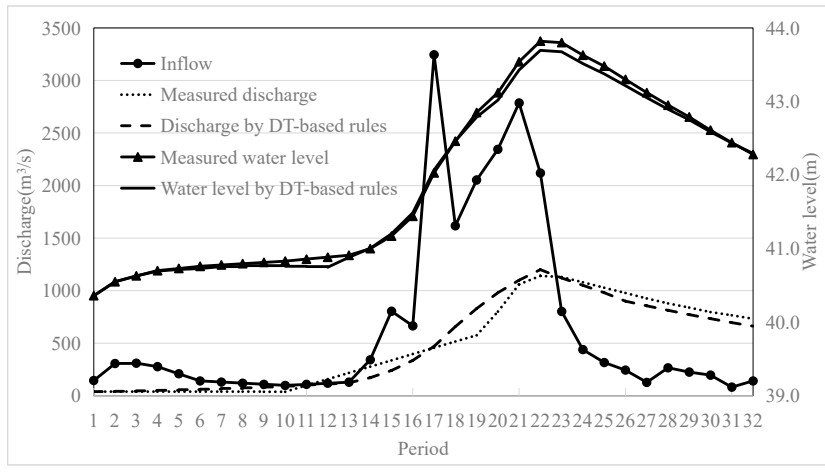

**Figure 3.** Operation hydrographs for 12 Aug 2019.

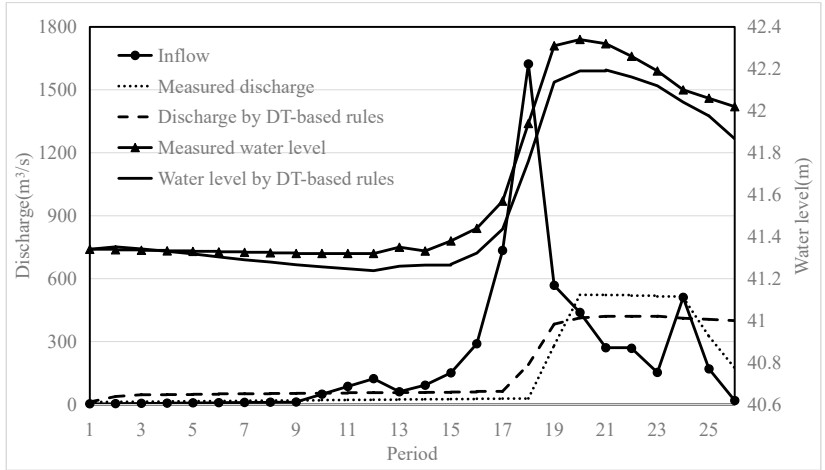

**Figure 4.** Operation hydrographs for 22 July 2020.

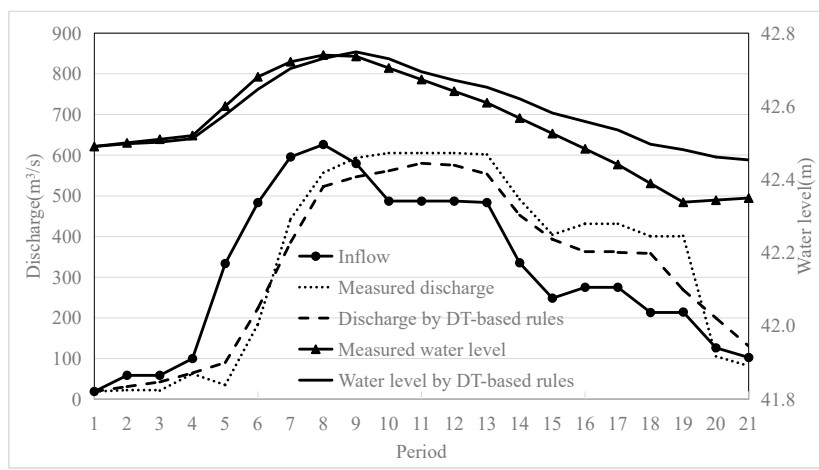

**Figure 5.** Operation hydrographs for 13 Aug 2021.

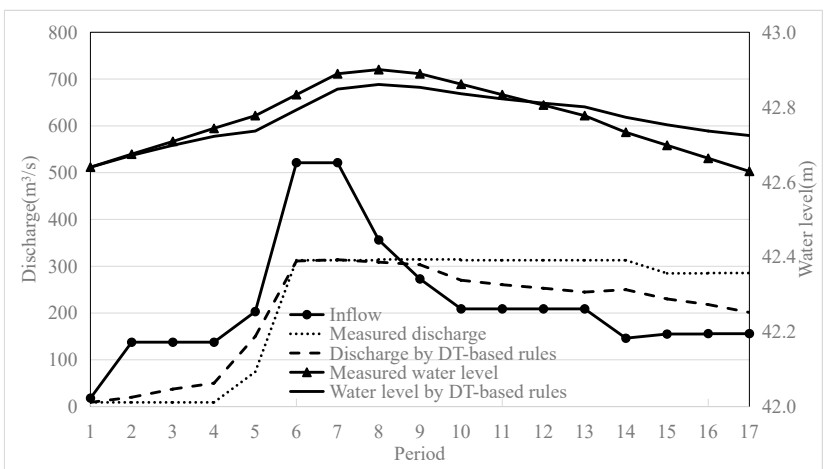

**Figure 6.** Operation hydrographs for 26 Aug 2021.

**Table 4.** Maximum discharges and water levels according to operating schemes.

| Items | Flood Number | ① DT-Based Rules | ② Measured Data | ③ Conventional Operation Rules | Relative Error between ① and ② (%) | Relative Error between ① and ③ (%) |
|---|---|---|---|---|---|---|
| Maximum discharge (m³/s) | 12 Aug 2019 | 1200 | 1140 | 1900 | 5.0 | 58.33 |
| | 22 July 2020 | 421 | 523 | 1000 | 24.23 | 137.53 |
| | 13 Aug 2021 | 580 | 605 | 1000 | 4.34 | 72.41 |
| | 26 Aug 2021 | 314 | 315 | 1000 | 0.32 | 218.46 |
| Maximum water level (m) | 12 Aug 2019 | 43.69 | 43.82 | 43.48 | 0.3 | 0.48 |
| | 22 July 2020 | 42.19 | 42.34 | 42.22 | 0.36 | 0.07 |
| | 13 Aug 2021 | 42.75 | 42.74 | 42.51 | 0.02 | 0.56 |
| | 26 Aug 2021 | 42.86 | 42.90 | 42.8 | 0.09 | 0.14 |

Note: 12 Aug 2019 is the flood number, which represents that the rainfall of the flood began on 12 August 2019.

The results shown in Table 4 are discussed below for each verification sample.

(1) For the 12 Aug 2019 flood, the maximum discharge of the operating scheme formulated using the conventional operation rules was the largest, followed by that of the operating scheme formulated using the DT-based rules, while the measured maximum discharge was the smallest; the relative error between ① and ② was small (5.0%), whereas the relative error between ① and ③ was large, reaching 58.33%. The measured maximum water level was the largest, followed by the maximum water level of the operating scheme formulated using the DT-based rules, while that of



the operating scheme formulated using the conventional operation rules was the smallest; the relative errors between ① and ② and between ① and ③ were both small (0.3% and 0.48%, respectively). It can be observed that the operating scheme formulated using the conventional operation rules was the best in terms of reservoir safety because its maximum water level was the smallest. However, this condition was the most unsafe downstream because the corresponding maximum discharge was the largest. The measured operating scheme was the best in terms of downstream safety because its maximum discharge was the smallest. However, this condition exhibited the highest maximum water level, indicating that it was the worst operating scheme in terms of reservoir safety. The maximum discharge and maximum water level provided by the operating scheme formulated using the DT-based rules were between those of the measured values and the operating scheme formulated using the conventional operation rules. At the same time, the maximum discharge of 1200 $m^3/s$ provided by the operating scheme formulated using the DT-based rules remained below the controlled discharge of 1900 $m^3/s$, indicating that it realized a suitable compromise and is therefore feasible.

(2) For the 22 July 2020 flood, the maximum discharge of the operating scheme formulated using the conventional operation rules was the largest, followed by the measured maximum discharge, while the maximum discharge of the operating scheme formulated using the DT-based rules was the smallest; the relative error between ① and ② was small (24.23%) whereas the relative error between ① and ③ was large, reaching 137.53%. The measured maximum water level was the largest, followed by the maximum water level of the operating scheme formulated using the conventional operation rules, while that of the operating scheme formulated using DT-based rules was the smallest; the relative errors between ① and ② and between ① and ③ were both small (0.36% and 0.07%, respectively). It can be observed that the operating scheme formulated using the DT-based rules was the best in terms of both reservoir safety and downstream safety because its water level and discharge were simultaneously the smallest. At the same time, the discharge of 421 $m^3/s$ dictated by the operating scheme formulated using the DT-based rules was less than the control discharge of 1000 $m^3/s$, confirming that this operating scheme was indeed the best among the three evaluated for this flood.

(3) For the 13 Aug 2021 flood, the maximum discharge of the operating scheme formulated using the conventional operation rules was the largest, followed by the measured maximum discharge, while the maximum discharge of the operating scheme formulated using the DT-based rules was the smallest; the relative error between ① and ② was small (4.34%), whereas the relative error between ① and ③ was large, reaching 72.41%. The maximum water level of the operating scheme formulated using the DT-based rules was the largest, followed by the measured maximum water level, while that of the operating scheme formulated using the conventional operation rules was the smallest; the relative errors between ① and ② and between ① and ③ were both small (0.02% and 0.56%, respectively). It can be observed that the operating scheme formulated using the conventional operation rules was the best in terms of reservoir safety because its maximum water level was the smallest. However, this condition was the most unsafe downstream because the corresponding maximum discharge was the largest. The operating scheme formulated using the DT-based rules was the best in terms of downstream safety because its maximum discharge was the smallest. However, this condition exhibited the highest maximum water level, indicating that it was the worst operating scheme in terms of reservoir safety. The difference between the maximum water level provided by the operating scheme formulated using the DT-based rules and the measured maximum water level is very small, that is 0.01 m, and the maximum discharge of 580 $m^3/s$ is below the controlled discharge of 1000 $m^3/s$, indicating that it is a feasible scheme.

(4) For the 26 Aug 2021 flood, the maximum discharge of the operating scheme formulated using the conventional operation rules was the largest, followed by the measured maximum discharge, while the maximum discharge of the operating scheme formulated using the DT-based rules was the smallest; the relative error between ① and ② was small (0.32%), whereas the relative error between ① and ③ was large, reaching 218.46%. The measured maximum water level was the largest, followed by the maximum water level of the operating scheme formulated using the DT-based rules, while that of the operating scheme formulated using the conventional operation rules was the smallest; the relative errors between ① and ② and between ① and ③ were both small (0.09% and 0.14%, respectively). It can be observed that the operating scheme formulated using the conventional operation rules was the best in terms of reservoir safety because its maximum water level was the smallest. However, this condition was the most unsafe downstream because the corresponding maximum discharge was the largest. The operating scheme formulated using the DT-based rules was the best in terms of downstream safety because its maximum discharge was the smallest. The maximum water level provided by the operating scheme formulated using the DT-based rules were between those of the measured values and the operating scheme formulated using the conventional operation rules. At the same time, the maximum discharge of 314 $m^3/s$ provided by the operating scheme formulated using the DT-based rules remained below the controlled discharge of 1000 $m^3/s$, indicating that it is a feasible scheme.

In summary, the operating scheme formulated using DT-based rules was shown to be feasible and, in some cases, better than the actual operating scheme and a scheme formulated using conventional operation rules.

## 4. Conclusions

In this paper, the DT algorithm was applied to formulate reservoir flood control operation rules that fully consider the influence of reservoir management experience, climate factors, and subsurface conditions of the watershed on discharge, realizing a fast and effective operating scheme that responds to various inflow scenarios under different hydrological periods. The following conclusions were obtained from this study:

(1) The C4.5 algorithm was used to construct DT-based flood control operation rules for a reservoir. This algorithm has the advantages of easy implementation and strong operability and fully considers the influence of the climate and underlying surface conditions of the watershed as well as the operating experience of management in the process of constructing an operating scheme.

(2) As can be seen from the results of the four verification samples, the maximum discharges of the operating schemes formulated using the DT-based rules with the flood number 22 July 2020, 13 Aug 2021, and 26 Aug 2021 are the smallest; the maximum discharge with the flood number 12 Aug 2019 is smaller than that of the operating scheme formulated using the conventional operation rules and only 5% larger than the measured maximum discharge. The maximum water levels of the operating schemes formulated using the DT-based rules with the flood numbers 12 Aug 2019 and 26 Aug 2021 are between those of the measured values and the operating scheme formulated using the conventional operation rules; the maximum water level with the flood number 22 July 2020 is the smallest; the maximum water level with the flood number 13 Aug 2021 is only 0.02% larger than the measured maximum value. To sum up, the operating scheme formulated using the DT-based rules was feasible and, in some cases, superior to the actual operating scheme and an operating scheme based on conventional operation rules. Among optimization algorithms, DT-based rules have the advantages of strong and convenient adaptability, allowing decision makers to guide real-time reservoir operation. Therefore, the DT-based method for constructing reservoir flood control operation rules proposed in this paper can provide practical guidance for the real-time operation of reservoirs.

(3) In this paper, only one reservoir is taken as an example, and future research should be popularized and applied in more reservoirs and reservoir group systems to verify the feasibility and effectiveness of this method.

**Author Contributions:** Conceptualization, Y.D. and Y.L.; methodology, Y.D. and H.W.; investigation, Y.D. and C.W.; data curation, C.W.; writing—original draft preparation, Y.D. and H.W.; writing—review and editing, Y.L.; funding acquisition, Y.D. and Y.L. All authors have read and agreed to the published version of the manuscript.

**Funding:** This research was funded by the Key Technology Research and Development Program of Shandong Province, grant number 2019GSF111043; the National Key Research and Development Program of China, grant number 2018YFC1508104; the National Natural Science Foundation of China, grant number 52079079; and the Natural Science Foundation of Jiangsu Province, grant number BK20191129.

**Institutional Review Board Statement:** Not applicable.

**Informed Consent Statement:** Not applicable.

**Data Availability Statement:** The datasets used and/or analyzed during the current study are available from the corresponding author upon reasonable request.

**Acknowledgments:** We express our deepest gratitude to the Rizhao Reservoir Management and Operation Center for their help in data support.

**Conflicts of Interest:** The authors declare no conflict of interest.

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
