# Peer review of "Construction and Application of Reservoir Flood Control Operation Rules Using the Decision Tree Algorithm"

_water, doi:10.3390/w13243654_

Round 1

Reviewer 1 Report

The authors did a good job of reviewing literature about reservoir operation rules and schemes. This complex computational arena is challenging due to so many different configurations, weather conditions, and operating scenarios, among other causes. The paper offers a good contribution with the literature review, but the development of the DT-based control algorithm is not very convincing. Reasons are that the arguments are focused mainly on the mathematics of the algorithm development and the presentation of the application and case study seems weak.  The conclusions are particularly weak.  The data set for training seems fairly good, but the testing on only two floods is disappointing.  Table 4, which shows data for conventional approaches, seems sketchy.

My conclusion is that the paper can be published on the basis of offering a novel proposal and continued good literature review. The paper would be much better if some explanations of the limited testing in comparison with conventional methods are presented and if the conclusions are strengthened.

Reviewer 2 Report

The manuscript is scientifically sound and can be published after minor revisions.

Page 2, Line 40: Did you not have more recent data than 2011?

Page 3, Lines 116-121: These can be deleted.

Figures 4 and 5: Use different line patterns.

Conclusions: Add an introduction to this section, and then present the results. Also, I recommend adding some text regarding future research recommendations.
